# Increased Prevalence of Headaches and Migraine in Patients with Psoriatic Arthritis and Axial Spondyloarthritis: Insights from an Italian Cohort Study

**DOI:** 10.3390/biomedicines12020371

**Published:** 2024-02-05

**Authors:** Annalisa Marino, Damiano Currado, Claudia Altamura, Marta Vomero, Onorina Berardicurti, Erika Corberi, Lyubomyra Kun, Andrea Pilato, Alice Biaggi, Irene Genovali, Pietro Bearzi, Marco Minerba, Antonio Orlando, Francesca Trunfio, Maria Quadrini, Chiara Salvolini, Letizia Pia Di Corcia, Francesca Saracino, Roberto Giacomelli, Luca Navarini

**Affiliations:** 1Rheumatology and Clinical Immunology, Department of Medicine, School of Medicine, University of Rome “Campus Bio-Medico”, 00128 Rome, Italy; a.marino@policlinicocampus.it (A.M.); d.currado@policlinicocampus.it (D.C.); m.vomero@unicampus.it (M.V.); erika.corberi@unicampus.it (E.C.); lyubomyra.kun@unicampus.it (L.K.); andrea.pilato@unicampus.it (A.P.); a.biaggi@unicampus.it (A.B.); irene.genovali@unicampus.it (I.G.); marco.minerba@unicampus.it (M.M.); antonio.orlando@unicampus.it (A.O.); francesca.trunfio@unicampus.it (F.T.); maria.quadrini@unicampus.it (M.Q.); chiara.salvolini@unicampus.it (C.S.); letiziapia.dicorcia@unicampus.it (L.P.D.C.); francesca.saracino@unicampus.it (F.S.); r.giacomelli@policlinicocampus.it (R.G.); l.navarini@policlinicocampus.it (L.N.); 2Clinical and Research Section of Rheumatology and Clinical Immunology, Fondazione Policlinico Universitario Campus Bio-Medico, 00128 Rome, Italy; 3Instituite of Neurology, Fondazione Policlinico Universitario Campus Bio-Medico, Via Alvaro del Portillo, 200, 00128 Rome, Italy; c.altamura@policlinicocampus.it; 4Unit of Headache and Neurosonology, Department of Medicine and Surgery, University of Rome “Campus Bio-Medico”, 00128 Rome, Italy

**Keywords:** headache, migraine, aura, psoriatic arthritis, axial spondyloarthritis

## Abstract

Background: Psoriatic arthritis (PsA) and axial spondyloarthritis (axSpA) are inflammatory diseases with shared genetic backgrounds and clinical comorbidities. Headache, a common global health issue, affects over 50% of adults and encompasses various types, including migraine, tension-type, and cluster headaches. Migraine, the most prevalent, recurrent, and disabling type, is often associated with other medical conditions such as depression, epilepsy, and psoriasis, but little is known about the relationship between autoimmune disease and the risk of migraine. Methods: A cross-sectional study was conducted from July to November 2022, enrolling 286 participants, including 216 with PsA, 70 with axSpA, and 87 healthy controls. Results: Headache prevalence was significantly higher in the PsA (39.81%) and axSpA (45.71%) patients compared to the healthy controls. The prevalence of migraine without aura was also significantly higher in both the PsA (18.52%) and axSpA (28.57%) groups compared to the healthy controls. Conclusions: These findings underscore the high burden of headache and migraine in PsA and axSpA participants, highlighting the need for improved management and treatment strategies for these patients.

## 1. Introduction

Psoriatic arthritis (PsA) and axial spondyloarthritis (axSpA) are inflammatory diseases involving the joints, characterized by shared genetic background as well as similar clinical manifestations and comorbidities [1]. Both PsA and axSpA are associated with decreased quality of life, loss of workability, and significant impairment of physical, psychological, and social functioning [2,3].

Headache is a common and often debilitating health problem, affecting over 50% of adults worldwide, according to the World Health Organization (WHO) [4]. Primary headaches may be classified into three main categories: migraine, tension-type, and cluster headaches. While episodic tension-type headache is the most frequent in population-based studies, cluster headache typically leads to significant disability, and a large number of these patients frequently need medical attention [5,6]. Lastly, the most common type is migraine, a chronic disease with an increasing prevalence, currently affecting more than 12% of the world’s population, and typically characterized by recurrent disabling attacks of headache and accompanying symptoms, including aura [7]. Headache, and more specifically migraine, is frequently associated with depression, epilepsy, stroke, and myocardial infarction [8], although a recent study also suggested an association with psoriasis. We may speculate that the endothelial dysfunction observed in psoriasis, together with episodes of local sterile meningeal inflammation, local hypersensitivity of pain pathways, and the release of the proinflammatory cytokines, involved in the pathogenesis of pso-riasis, may promote migraine, by sensitization of meningeal nociceptors and peripheral nerve endings, via the activation of mitogen-activated p38 protein kinases and increased dural mechano-sensitivity [9,10,11,12].

Several studies showed a close relationship between autoimmune disease and the risk of migraine, particularly in multiple sclerosis and inflammatory bowel disease [13,14].

As far as rheumatological diseases are concerned, previous studies have focused mainly on cohorts of patients suffering from psoriasis or on cohorts of patients with neuropathic pain and chronic inflammatory diseases, to investigate the physiopathology of pain in chronic inflammatory rheumatic diseases (CIRDs), showing that the highest prevalence of migraines was observed in patients with PsA, while the highest prevalence of neuropathic pain was found among patients with spondyloarthritis (SpA) [15,16,17].

In this setting, our study aimed to evaluate the prevalence of headache and of migraine with and without aura, in a cohort of patients with PsA and axSpA followed in a single Italian rheumatologic center and to correlated this datum with the demographic and clinical data of the patients. To the best of our knowledge, this represents the first study endeavoring to examine the prevalence of headaches and migraines specifically in Italian participants diagnosed with PsA and axSpA.

## 2. Materials and Methods

A single-center, cross-sectional, observational cohort study was conducted on 216 PsA patients and 70 axSpA patients. At baseline, PsA participants who fulfilled the Classification Criteria for Psoriatic Arthritis (CASPAR) [18] and who were undergoing at least 6 months of follow-up treatment with conventional and/or biologic disease-modifying antirheumatic drugs were considered potentially eligible for the study. For the axSpA group, we used the New York criteria for ankylosing spondylitis or the axial Assessment of SpondyloArthritis International Society (ASAS) criteria for non-radiographic spondyloarthropathy [19]. Eighty-seven consecutive healthy controls were enrolled at the Rheumatology Unit, Campus Bio-Medico University, Rome, Italy, from July 2022 to November 2022. The study was approved by the Ethics Committee of the University Campus Bio-Medico of Rome and conducted in conformity with the Declaration of Helsinki and its later amendments. The following inclusion criteria were considered: both sex, age 18–80 years, and the fulfilment of CASPAR and/or ASAS criteria. History of any psychiatric disorder according to DSM-V prior to recruitment, history of any malignancy, pregnancy, or inability to express informed consent to participate in the study were considered as exclusion criteria. Clinical assessment encompassed the number of tender joints (of the 68 assessed joints) and swollen joints (of the 66 assessed joints), enthesitis, and dactylitis. Enthesitis was assessed using the Leeds Enthesitis Index (LEI), and dactylitis was assessed as present or absent. Skin assessment was performed using the Psoriasis Area Severity Index (PASI). At enrolment, the following disease activity scores were collected: Disease Activity for PSoriatic Arthritis (DAPSA), Composite Psoriatic Disease Activity Index (CPDAI), minimal disease activity (MDA) and very low disease activity (VLDA), and the Bath Ankylosing Spondylitis Disease Activity Index (BASDAI). Patients fulfilling the 2016 American College of Rheumatology revised diagnostic criteria were classified as affected by concomitant fibromyalgia [20]. Furthermore, the following axSpA disease activity scores were collected: BASDAI, Bath Ankylosing Spondylitis Functional Index (BASFI), and the Ankylosing Spondylitis Disease Activity Score-C-Reactive Protein (ASDAS-CRP).

Headache and migraine diagnosis were assigned using the international classification of headache disorders (ICHD), after completion of a semi-structured diagnostic interview [21]. The global question assessing migraine was formulated as the following: “Have you had a headache in the last three months?” and the yes/no response was used as a dichotomic variable. After that question, the screen employed three questions concerning migraine: “During the last three months, did you have the following with your headaches? Photophobia: Did light bother you (a lot more than when you do not have headaches)? Incapacity: Did your headaches limit your ability to work, study, or do what you needed to do for at least one day? Nausea: Did you feel nauseated or sick to your stomach?” In order to better recognize migraine aura, the Visual Aura Rating Scale (VARS) was developed. The VARS score comprises five major symptoms that are individually weighted: (1) duration: 5–60 min (3 points); (2) gradual development ≥ 5 min (2 points); (3) presence of scotoma (2 points); (4) presence of zig-zag lines (2 points); and (5) unilateral presentation (1 point). A summation score ≥ 5 demonstrates a sensitivity of 91–96% and a specificity of 96–98% for migraine aura [22,23].

Continuous data are described as median (25–75th Pctl), whilst categorical variables are described as percentages (%). The Shapiro–Wilk test was used to evaluate the normality of data. χ^2^ was used for the analysis of contingency tables. The whole statistical analysis was performed using Stata v.14. *p*-values < 0.05 were considered significant.

## 3. Results

Two hundred and eighty-six participants (216 participants with PsA and 70 with axSpA) and 87 healthy controls were included in the study. The main demographic, anthropometric, and clinical characteristics of the study population are reported in Table 1 for PsA and Table 2 for axSpA.

The patients with PsA and headache were predominantly female [*n* = 66 (76.74%) vs. *n* = 20 (23.26%), *p* = 0.004), had a higher comorbidity burden [Charlson Comorbidity Index 2 (1–3) vs. 1 (0–3), *p* = 0.02], and had a higher prevalence of enthesitis (59.76% vs. 45.24%, *p* = 0.02). They also had higher use of nonsteroidal anti-inflammatory drugs (44.19% vs. 28.46%, *p* = 0.01), a higher number of tender joints [TJ 2.5 (1–5) vs. 1 (0–4), *p* = 0.01] and swollen joints [SJ 0 (0–1) vs. 0 (0–0), *p* = 0.02], higher Leeds Enthesitis Index scores [LEI 0 (0–1) vs. 0 (0–0), *p* = 0.004], and higher Health Assessment Questionnaire (HAQ) scores [1 (0.63–1.75) vs. 0.75 (0.25–1.33), *p* = 0.008] than those without headache.

Among the axSpA patients, those with headache were also predominantly female [*n* = 21 (75%) vs. *n* = 16 (42.11%), *p* = 0.005] and had a higher prevalence of concomitant fibromyalgia (56.25% vs. 21.05%, *p* = 0.003), higher frequency of HLA-B27 (39.47% vs. 15.62%, *p* = 0.025), and a higher rate of biological drug usage, in particular with the use of anti-TNF-alpha therapy (39.47% vs. 18.75%, *p* = 0.015).

Participants in the PsA and axSpA groups were comparable and did not differ at baseline from the group of healthy controls with regard to gender, Charlson comorbidity index [24], or prevalence of Body Mass Index (BMI) (Appendix A).

All the participants were Caucasian, with a large preponderance of females (65.74% in PsA, 57.14% in SA, and 63.22% in healthy controls) and a median age of 57 years in both disease groups. The average duration of disease was 102 months for PsA and 117.5 for axSpA patients. The average BMI was 26.5 for PsA and 25.76 for axSpA. About 70% of both PsA and axSpA patients were receiving biological therapy.

Among the 216 patients with PsA, 86 (39.81%) reported headache, whilst 32 (45.71%) out of the 70 axSpA participants reported headache.

We analyzed the prevalence of headache, and of migraine with and without aura, in our patient population and compared it with the prevalence in the control population.

In the patients with PsA, the prevalence of headache was 39.81%, which was significantly higher than the prevalence of 26.44% in the healthy controls (*p* = 0.028). The prevalence of migraine without aura was 18.52% in the patients with PsA, compared to 9.2% in the healthy controls (*p* = 0.044); lastly, the prevalence of aura was not significantly different between the patients with PsA and the healthy controls (Table 3).

Among the patients with axSpA, the prevalence of headache was 45.71%, and the prevalence of migraine without aura was 28.57%, both of which were significantly higher than the respective prevalences in healthy controls (*p* < 0.05). The prevalence of aura was not significantly different between the patients with axSpA and the healthy controls (Table 4).

Considering the two patient populations together, the prevalence of headache (41.26%) and the prevalence of migraine without aura (20.98%) were significantly higher than in the healthy controls (*p* = 0.013) (Table 5).

## 4. Discussion

This study showed a higher prevalence of headache and migraine without aura in the patients with PsA and axSpA compared to the healthy controls. This result suggests a link between the systemic inflammation of spondyloarthritis and the immunological and pain-related patterns underlying the development of headache and migraine [25,26]. This is further confirmed by other studies showing similar findings in the case of autoimmune diseases, particularly in those characterized by chronic pain, for instance, multiple sclerosis [27,28].

In our cohort, patients with headache did not show higher levels of pain, evaluated using patient pain (PP), a tool used by clinicians to evaluate, through a visual analogue scale, the self-reported pain of arthritis patients. However, within the PsA patients, the incidence of headaches shows a correlation with an increased count of tender and swollen joints, along with higher scores in the LEI and HAQ. This indicates that individuals experiencing a more pronounced burden of PsA symptoms are more inclined to develop headaches.

Furthermore, the PsA patients experiencing headaches exhibited a notably elevated comorbidity index compared to those without headaches. This result has been recently confirmed in inflammatory bowel diseases, which may be considered pathogenetically linked to PsA [29].

As far as axSpA is concerned, we showed that the patients who experienced headache used anti-TNF-alpha therapy more frequently. This result may be linked to a more severe disease activity in axSpA patients developing headache compared to those without headache, thus confirming that chronic inflammation may be linked to the onset of headache.

A noteworthy predominance of females was observed among the PsA and axSpA patients with comorbid headache. This phenomenon is rooted in the documented estrogen-dependent nature of headache; it is widely recognized that variations in estrogen levels, whether increased or decreased, can act as triggers for headaches, including migraines, during different stages of reproductive age such as menstruation, pregnancy, or menopause [30].

In fact, migraine is considered a “brain state,” with various alterations in brain network activity being influenced by hormonal fluctuations. Estrogen, in particular, has been linked to migraines since the 1970s, and studies have shown that migraines most commonly occur during a rapid decline in estrogen levels in the late luteal phase and early follicular phase, known as the hypothesis of estrogen withdrawal [31].

Although withdrawing progesterone does not directly impact migraine onset, it is believed to influence the headache-promoting effects of estrogen fluctuations. The interplay between these hormones and the reduction in GABAergic activity stemming from progesterone decrease in the late luteal phase might be pivotal in comprehending migraine pathophysiology. Moreover, cyclical hormonal variations modulate calcitonin gene-related peptide (CGRP) levels, which are elevated in women of reproductive age, especially during heightened estrogen phases [30,32].

Both spondyloarthritis and headache show an increased risk of cardiovascular events [33,34,35]. It has been demonstrated that the persistence of increased levels of C-reactive protein and high disease activity are predictors of cardiovascular disease in patients with axial spondyloarthritis [33,34]. On the other hand, migraine has been associated with an unfavorable cardiovascular risk profile and a 2-fold increase in the 10-year risk of coronary heart disease [36].

The common pathogenesis of arthritis and migraine may also be related to the vascular and neurogenic components. In fact, in murine models of polyarthritis, the density of CGRP-IR fibers is significantly increased and contributes to the generation and maintenance of arthritis pain [37]. Migraine has been associated with an unfavorable cardiovascular risk profile and a 2-fold increase in the 10-year risk of coronary heart disease. At the same time, CGRP and its canonical receptor, a heterodimer consisting of a calcitonin-like receptor and a protein 1 that modulates receptor activity, are widely expressed in both the central and peripheral nervous systems, as well as in the trigeminal vascular system. They seem to trigger vasodilation and migraine, as demonstrated by plasma levels of CGRP that positively correlate with the intensity and timing of headaches, and intravenous infusion of CGRP that causes migraine-like symptoms in patients with migraine [38,39,40].

The common cytokine-pattern background could explain the genesis and perpetuation of this process, as cytokines are important endogenous substances involved in the immune and inflammatory responses. Further longitudinal research is required to investigate the role of the spondyloarthritis–migraine comorbidity in defining patients at a very high risk of cardiovascular events.

Regarding the use of biological drugs, our findings indicate that among the patients with axSpA encountering headaches, there was a higher usage of anti-TNF-alpha therapy compared to the patients without headaches. This result, which may seem counterintuitive, could be associated with heightened disease activity (leading to the administration of bDMARDs) in the axSpA patients experiencing headaches [41].

In addition, it is understood that there exist two variations of TNF-alpha: a soluble variant that primarily interacts with the TNF type 1 receptor (TNFR1), and a transmembrane variant that interacts with the TNF type 2 receptor (TNFR2). Activation of TNFR1 leads to apoptosis and persistent inflammation; conversely, TNFR2 activation supports cell survival, resolves inflammation, and triggers remyelination. TNFR2 is notably prevalent in the central nervous system; thus, the use of TNF-alpha inhibitors might hinder the anti-inflammatory and regenerative impacts of the transmembrane TNF-alpha on TNFR2, potentially fostering demyelination and possibly triggering migraines in patients utilizing such inhibitors [42].

Interpreting our present findings involves acknowledging various limitations and strengths. The observational design of our study impeded a definitive establishment of causality. The determination of whether the identified effect was influenced, in part, by mild systemic inflammation stemming from arthritis or predominantly arose due to shared lifestyle aspects and alternative mechanisms necessitates deeper investigation. Additionally, we could not distinguish whether the incidence of headache and migraine preceded the development of arthritis or not. However, the association between spondyloarthritis and headache and migraine observed in our study is supported by the fact that the prevalence of migraine in our population is nearly twice that usually observed in age- and sex-matched healthy individuals. We are aware of the high prevalence of fibromyalgia in both populations, likely due to the fact that we used 2016 ACR diagnostic criteria and not clinical parameters, such as tender points or self-reported diagnosis. The presence of concurrent fibromyalgia, especially in the context of the headache symptom, which is a hallmark symptom of fibromyalgia, is certainly a limiting factor in our study. Other previous studies have demonstrated the lack of an association between headache and comorbid fibromyalgia in rheumatoid arthritis and spondyloarthritis; however, in these studies, the prevalence of fibromyalgia appeared to be underestimated, as it was solely defined via a self-reported diagnosis. We have chosen not to exclude patients with comorbid fibromyalgia from our analysis in order to fully reflect the characteristics of our real-world population, in which fibromyalgia is one of the most prevalent comorbidities in spondyloarthritis.

In relation to the AxSpA patient cohort, our study did not specifically examine cervical involvement, such as cervical spondylosis or ankylosis, despite the potential for these conditions to trigger headaches. Our assessment did not isolate spinal involvement in distinct regions but rather evaluated it comprehensively. At the same time, the low number of participants potentially does not fully capture the heterogeneity of the disease. Lastly, it is essential to note that our study comprised predominantly individuals of Caucasian descent. Therefore, caution must be exercised when generalizing these findings to populations of different ethnicities, considering the potential variations in migraine prevalence between Caucasians and non-Caucasian groups.

In conclusion, our study provides a “real-life” estimation of the prevalence of headache and migraine in the spondyloarthritis population, finding an increased risk of headache and migraine in patients with PsA and axSpA. These findings should open a new avenue for increasing awareness of the importance of examining the risk of migraine and headache in rheumatologic patients, correlating it with other variables such as an increased cardiovascular risk, and developing effective therapies to treat both conditions.

## Figures and Tables

**Table 1 biomedicines-12-00371-t001:** PsA patients demography and characteristics; comparisons between patients with and without headache. PsA: Psoriatic Arthritis; BMI: body mass index; IBD: inflammatory bowel diseases; DMARD: disease-modifying anti-rheumatic drug; csDMARD: conventional synthetic DMARD; bDMARD: biologic DMARD; tsDMARD: targeted synthetic DMARD; GCC: glucocorticoids; NSAID: non-steroidal anti-inflammatory drug; TJ: tender joints; SJ: swollen joints; PP: patient pain; PtGA: patient global assessment; LEI: Leeds enthesitis index; CRP: C-reactive protein; ESR: erythrocyte sedimentation rate; HAQ: Health Assessment Questionnaire; PASI: Psoriasis area severity index; DAPSA: Disease Activity in Psoriatic Arthritis; MDA: minimal disease activity; VLDA: very low disease activity.

Variables	Entire PsA Population 216 (100%)	Headache = 1 86 (39.81%)	Headache = 0 130 (60.19%)	*p*
Age (years)	57 (49–63)	56 (48–62)	58 (50–64)	0.36
Female, *n* (%)	142 (65.74%)	66 (76.74%)	20 (23.26%)	**0.004**
Disease duration (months)	102 (53–144)	89 (48–140)	108 (60–150)	0.15
BMI	26.5 (23.68–29.9)	27.15 (24.1–29.9)	26.2 (23.5–30.2)	0.54
Charlson Comorbidity Index	2 (1–3)	2 (1–3)	1 (0–3)	**0.02**
Fibromyalgia, *n* (%)	76 (35.68%)	33 (38.82%)	43 (33.59%)	0.2
Peripheral arthritis, *n* (%)	211 (97.69%)	84 (97.67%)	127 (97.69%)	0.6
Axial involvement (%)	118 (55.66%)	48 (56.47%)	70 (55.12%)	0.47
Enthesitis (%)	106 (50.96%)	49 (59.76%)	57 (45.24%)	**0.02**
Dactylitis (%)	43 (20.28%)	21 (25.30%)	22 (17.05%)	0.1
Psoriasis (%)	141 (65.89%)	52 (61.90%)	89 (68.46%)	0.2
Nail psoriasis (%)	52 (24.30%)	20 (23.53%)	32 (24.81%)	0.48
Uveitis (%)	10 (4.69%)	5 (5.88%)	5 (3.91%)	0.36
IBD	10 (4.69%)	4 (4.76%)	6 (4.65%)	0.6
DMARDs no use, *n*(%)	104 (48.15%)	42 (48.84%)	62 (47.69%)	0.33
csDMARDs duration therapy, months	5 (0–36)	4 (0–30)	5 (0–40)	0.98
b/tsDMARDs no use, (%) Infliximab Adalimumab Etanercept Golimumab Certolizumab-pegol Secukinumab Ixekizumab Ustekinumab Apremilast	66 (30.56%) 3 (1.39%) 61 (28.24%) 29 (13.43%) 13 (6.02%) 3 (1.39%) 9 (4.17%) 7 (3.24%) 18 (8.33%) 7 (3.24%)	27 (31.40%) 3 (3.49%) 31 (36.05%) 11 (12.79%) 2 (2.33%) 3 (3.49%) 2 (2.33%) 3 (3.49%) 2 (2.33%) 2 (2.33%)	39 (30%) 0.00 30 (23.08%) 18 (13.85%) 11 (8.46%) 0.00 7 (5.38%) 4 (3.08%) 16 (12.31%) 5 (3.85%)	0.47
bDMARDs duration therapy, months	13.5 (0–45)	8 (0–40)	18 (0–45)	0.28
GCC (%)	17.13%	19.77%	15.38%	0.25
NSAIDs (%)	34.72%	44.19%	28.46%	**0.01**
TJ	2 (0–5)	2.5 (1–5)	1 (0–4)	**0.01**
SJ	0 (0–1)	0 (0–1)	0 (0–0)	**0.02**
PP	6 (2–8)	6 (3–8)	5 (2–8)	0.39
PtGA	6 (3–7)	4 (4–7.5)	5 (2–7)	0.41
LEI	0 (0–0)	0 (0–1)	0 (0–0)	**0.004**
Dactilitis	1.36%	1.45%	1.22%	0.3
CRP mg/dL	0.3 (0.17–0.6)	0.3 (0.17–0.7)	0.3 (0.17–0.5)	0.72
ESR	13 (7.5–21.5)	14 (7–22)	13 (8–21)	0.51
HAQ	0.88 (0.33–1.5)	1 (0.63–1.75)	0.75 (0.25–1.33)	**0.008**
PASI	0 (0–0)	0 (0–0)	0 (0–0.4)	0.47
DAPSA	14.34 (7.1–20.1)	16.08 (8.67–21)	14 (7–19.4)	0.17
MDA	22.07%	23.26%	21.26%	0.42
VLDA	14.55%	13.95%	14.96%	0.5
Migraine	40 (18.52%)	40 (46.51%)	0	
Aura	6 (2.78%)	6 (6.98%)	0	

**Table 2 biomedicines-12-00371-t002:** AxSpA patients demography and characteristics; comparisons between patients with and without headache. axSpA: axial spondyloarthritis; BMI: body mass index; IBD: inflammatory bowel diseases; DMARD: disease-modifying anti-rheumatic drug; csDMARD: conventional synthetic DMARD; bDMARD: biologic DMARD; GCC: glucocorticoids; NSAID: non-steroidal anti-inflammatory drug; PP: patient pain; PtGA: patient global assessment; CRP: C-reactive protein; ESR: erythrocyte sedimentation rate; BASFI: Bath Ankylosing Spondylitis Functional Index; BASDAI: Bath Ankylosing Spondylitis Disease Activity Index; ASDAS-CRP: Ankylosing Spondylitis Disease Activity Score with CRP.

Variables	Entire axSpA Population 70 (100%)	Headache = 1 32 (45.71%)	Headache = 0 38 (54.29%)	*p*
Age (years)	57 (48–66)	54.5 (46–62)	61 (52–70)	0.12
Female, *n* (%)	40 (57.14%)	21 (75%)	16 (42.11%)	**0.005**
Disease duration (months)	117.5 (65–156)	120 (63–148.5)	115 (65–156)	0.99
BMI	25.76 (23.8–28.05)	25.78 (23.35–28.05)	25.1 (24.33–28.91)	0.64
Charlson Comorbidity Index	2 (1–3)	2 (1–3)	3 (1–3)	0.59
Fibromyalgia, (%)	26 (37.14%)	18 (56.25%)	8 (21.05%)	**0.003**
HLA-b27, %	20 (28.57%)	5 (15.62%)	15 (39.47%)	**0.025**
Peripheral arthritis (%)	48 (68.57%)	28 (87.50%)	20 (52.63%)	**0.002**
Enthesitis (%)	21 (52.63%)	12 (40.00%)	9 (25.00%)	0.15
Dactylitis (%)	13 (20%)	4 (13.33%)	9 (25.71%)	0.17
Uveitis (%)	13 (18.57%)	5 (15.62%)	8 (21.05%)	0.39
IBD	13 (18.57%)	3 (9.38%)	10 (26.32%)	0.06
csDMARDs no use,	58 (82.86%)	25 (78.12%)	33 (86.84%)	0.64
b/tsDMARDs no use, (%) Infliximab Adalimumab Etanercept Golimumab Certolizumab-pegol Secukinumab	21 (30%) 5(7.14%) 17 (24.29%) 4 (5.71%) 11 (15.71%) 3 (4.29%) 9 (12.86%)	6 (18.75%) 5(15.62%) 5 (15.62%) 2 (6.25%) 5 (15.62%) 2 (6.25%) 7 (21.88%)	15 (39.47%) 0.00 12 (31.58%) 2 (5.26%) 6 (15.79%) 1 (2.63%) 2 (5.26%)	**0.015**
bDMARDs duration therapy, months	20 (0–58)	37.5 (2–60)	18 (0–50)	0.36
GCC (%)	18.75%	15.71%	13.16%	0.37
NSAIDs (%)	25.71%	18.75%	31.58%	0.17
PP	6.5 (3–8)	7 (4.75–8)	5 (3–8)	0.6
PtGA	6.5 (3–8)	7 (5–8)	5.5 (2–7.5)	0.06
LEI	0 (0–0)	0 (0–0)	0 (0–0)	0.86
CRP mg/dL	0.29 (0.2–0.42)	0.2 (0.1–0.47)	0.3 (0.2–0.42)	0.38
ESR	13 (6–22)	15 (8–26)	10.5 (5–19)	0.15
BASFI	4.1 (1.66–6)	4.5 (2–6)	3.6 (1.4–6)	0.37
BASDAI	4.6 (2.9–6)	5.125 (3.85–6.7)	4 (2.9–5.4)	0.07
ASDAS-CRP	2.87 (1.97–3.15)	3.05 (2.37–3.15)	2.48 (1.7–3.26)	0.26
Migraine	20 (28.57%)	20 (62.50%)	0	
Aura	8 (11.43%)	28 (25%)	0	

**Table 3 biomedicines-12-00371-t003:** Prevalence of Headache, Migraine, and Aura in the PsA patients.

	PsA (*n* = 216)	Healthy Controls (*n* = 87)	*p*
Headache	39.81%	26.44%	**0.028**
Migraine	18.52%	9.2%	**0.044**
Aura	15%	26.09%	0.3

**Table 4 biomedicines-12-00371-t004:** Prevalence of Headache, Migraine, and Aura in the axSpA patients.

	axSpA (*n* = 70)	Healthy Controls (*n* = 87)	*p*
Headache	45.71%	26.44%	**0.012**
Migraine	28.57%	9.2%	**0.02**
Aura	40%	26.09%	0.3

**Table 5 biomedicines-12-00371-t005:** Prevalence of Headache, Migraine, and Aura in the entire population.

	Entire Population (PsA + axSpA) (*n* = 286)	Healthy Controls (*n* = 87)	*p*
Headache	41.26%	26.44%	**0.013**
Migraine	20.98%	9.2%	**0.013**
Aura	23.33%	26.09%	0.793

## Data Availability

The raw data supporting the conclusions of this article will be made available by the authors on request.

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
