# Peer review of "Increased Prevalence of Headaches and Migraine in Patients with Psoriatic Arthritis and Axial Spondyloarthritis: Insights from an Italian Cohort Study"

_biomedicines, 2024, doi:10.3390/biomedicines12020371_

Round 1

Reviewer 1 Report

Comments and Suggestions for Authors

I read your manuscript with interest. Although the numbers were small, the main concern is about the methodology. It is unclear whether the authors used the international classification of headache disorders (ICHD) definition of migraine.

1. In the methodology, authors suggested they asked the patients ‘’Have you had a headache in the last three months". This could have led to patients reporting any headache, as the question seemed vague. The data suggests that a good proportion of the patients had headaches without visual aura.

2. It appears more patients in the AxSpA group reported headaches than psoriatic arthritis. Could this be because AxSpa patients might have had cervical spinal involvement? Did the authors exclude patients with cervical spondylosis or ankylosis from axial spondylarthritis, which can also cause headaches?

3. It was unclear whether the authors excluded patients with chronic headaches.

4. Authors have mentioned among the AxSpA patients, those with headaches had a higher prevalence of concomitant fibromyalgia. Could it be possible that the headache in AxSpA patients is perhaps related to fibromyalgia?

5. I suggest authors discuss the prevalence of migraines in patients with other autoimmune rheumatic conditions. That would be a better comparative data than using multiple sclerosis example.

6. In Table 1, please insert numbers against peripheral arthritis to uveitis. Only proportion data were provided.

7. In Table 1, please provide full forms, for example, PASI. Please ensure reporting all abbreviations for the benefit of non-rheumatology readers. These should be placed in the footnotes rather than the top. There are similar issues with a lack of uniformity in reporting the data.

8. Line 30- when using the abbreviations, please present them along with full form before using them separately.

Author Response

Dear Reviewer,

I sincerely thank you for the comments. We have followed your suggestions and amended our manuscript accordingly. Please find below our responses to the queries raised. We trust these have satisfied your concerns and look forward to hearing from you.

  1. It is unclear whether the authors used the international classification of headache disorders (ICHD) definition of migraine. We thank the Reviewer for this suggestion. We modified this sentence in the method section: Headache and migraine diagnosis were assigned using the international classification of headache disorders (ICHD), after completing a semi-structured diagnostic interview.
  2. In the methodology, authors suggested they asked the patients ‘’Have you had a headache in the last three months". This could have led to patients reporting any headache, as the question seemed vague. The data suggests that a good proportion of the patients had headaches without visual aura. We thank the Reviewer for this question. As specified in the methods, the headache and migraine questionnaire were scored based on the International Headache Society criteria after completing a semi-structured diagnostic interview. Indeed, this study demonstrates that the three-question ID Migraine Screener test, consisting of questions on disability, nausea and photophobia, is a valid and reliable screening tool for migraine in primary care settings. As reported in the study included in the reference (Lipton RB et al, Neurology 2003 Aug 12) this setting had a sensitivity of 0.81 (95% CI, 0.77 to 0.85) and a specificity of 0.75 ( 95% CI, 0.64 to 0.84), relative to an IHS-based migraine diagnosis assigned by a headache specialist. We reported these sentences in the method section: Headache and migraine diagnosis were assigned based on International Headache Society criteria after completing a semi-structured diagnostic interview. The global question assessing migraine was formulated as the following: ‘’Have you had a headache in the last three months" and the yes/no response was used as a dichotomic variable. After that question, for migraine the screen employs three questions: “During the last three months, did you have the following with your headaches? Photophobia: Did light bother you (a lot more than when you do not have headaches)? Incapacity: Did your headaches limit your ability to work, study, or do what you needed to do for at least one day? Nausea: Did you feel nauseated or sick to your stomach? In order to better recognize migraine aura, the Visual Aura Rating Scale (VARS) was developed. The VARS score is comprised of five major symptoms that are individually weighted: 1) duration: 5–60 minutes (3 points); 2) gradual development ≥ 5 minutes (2 points); presence of scotoma (2 points); presence of zig-zag lines (2 points); and unilateral presentation (1 point). A summation score ≥ 5 demonstrates a sensitivity of 91–96% and a specificity of 96–98% for migraine aura.
  3. It appears more patients in the AxSpA group reported headaches than psoriatic arthritis. Could this be because AxSpa patients might have had cervical spinal involvement? Did the authors exclude patients with cervical spondylosis or ankylosis from axial spondylarthritis, which can also cause headaches? We did not consider in our group of AxSpA patients those with involvement of the cervical spine or those with cervical spondylosis or ankylosis, as the clinical data relating to the involvement of the different districts were not always available. We arrived at the latter as a limitation of the study from line 308: “As long as AxSpA group of patients, we did not investigate the cervical involvement (in terms of cervical spondylosis or ankylosis), even if this could cause headaches.
  4. It was unclear whether the authors excluded patients with chronic headaches. Thanks for the suggestion. We did not exclude patients with chronic headaches, who were included in the study. To respond to this suggestion, we inserted in the text, as a limitation of the study, at line 308: "Additionally, we could not distinguish between the incidence of headache and migraine, whether it preceded the development of arthritis or not" Furthermore, we compared the PsA and AxSpA population with a population of healthy controls and even for the healthy controls we did not exclude participants with chronic headaches. Furthermore, as reported in method, the question “Have you had a headache in the last three months?” aimed to evaluate episode of acute acute headaches (during the last three months) even in patients with chronic ones.
  5. Authors have mentioned among the AxSpA patients, those with headaches had a higher prevalence of concomitant fibromyalgia. Could it be possible that the headache in AxSpA patients is perhaps related to fibromyalgia? Really thank to highlight this topic. We added the following sentences as limitations in the discussion section: We are aware of the high prevalence of fibromyalgia in both populations, likely due to the fact that we used 2016 ACR diagnostic criteria and not clinical parameters such as tender points or self-reported diagnosis. The presence of concurrent fibromyalgia, especially in the context of the headache symptom, which is a hallmark symptom of fibromyalgia, is certainly a limiting factor in our study. Other previous studies have demonstrated the lack of association between comorbid fibromyalgia in rheumatoid arthritis and spondyloarthritis and headache; however, in these studies, the prevalence of fibromyalgia appeared to be underestimated, as it was solely defined as a self-reported diagnosis. We have chosen not to exclude patients with comorbid fibromyalgia from our analysis in order to fully reflect the characteristics of our real-world population, in which fibromyalgia is one of the most prevalent comorbidities in spondyloarthritis.
  6. I suggest authors discuss the prevalence of migraines in patients with other autoimmune rheumatic conditions. That would be a better comparative data than using multiple sclerosis example. Thanks for the suggestion, we have added in the introduction the data available so far on the cohort of patients with rheumatological diseases, from line 71: " As far as rheumatological diseases are concerned, previous studies have focused main-ly on cohorts of patients suffering from psoriasis or on cohorts of patients with neuropathic pain and chronic inflammatory diseases, to investigate the physiopathology of pain in chronic inflammatory rheumatic diseases (CIRDs), showing that the highest prevalence of migraines was observed in patients with PsA, while the highest prevalence of neuropathic pain was found among patients with Spondyloarthritis (SpA)." and in the discussion a link on the pathogenesis and related cytokine profile, from line 278 "The common pathogenesis of arthritis and migraine may also be related to the vascular and neurogenic components. In fact, it has been demonstrated that the persistence of increased levels of C-reactive protein and high disease activity are predictors of cardio-vascular disease in patients with axial spondyloarthritis. Additionally, in murine models of polyarthritis, the density of CGRP-IR fibers is significantly increased and contributes to the generation and maintenance of arthritis pain. Migraine has been associated with an unfavorable cardiovascular risk profile and a 2-fold increased 10-year risk of coronary heart disease.At the same time, CGRP and its canonical receptor, a heterodimer consisting of a calcitonin-like receptor and a protein 1 that modulates receptor activity, are widely expressed in both the central and peripheral nervous systems, as well as in the trigemino-vascular system.They seem to trigger vasodilation and migration, as demonstrated by plasma levels of CGRP that positively correlate with the intensity and timing of headaches, and intravenous infusion of CGRP that causes migraine-like symptoms in patients with migraine. The common cytokine pattern background could explain the genesis and perpetuation of this process, as cytokines are important endogenous substances involved in immune and inflammatory responses."
  7. In Table 1, please insert numbers against peripheral arthritis to uveitis. Only proportion data were provided. We apologize for this typo. We reported numbers, both for PsA and axSPA table, from peripheral arthritis to IBD.
  8. In Table 1, please provide full forms, for example, PASI. Please ensure reporting all abbreviations for the benefit of non-rheumatology readers. These should be placed in the footnotes rather than the top. There are similar issues with a lack of uniformity in reporting the data. Thank you for the suggestions; we agree with the reviewer that the abbreviations used can often be confusing, however they have been inserted at the top of the table as per the formatting indication according to Biomedicines.
  9. Line 30- when using the abbreviations, please present them along with full form before using them separately. We apologize for this typo. we have added the full form before (line 23: Psoriatic arthritis (PsA) and Axial Spondyloarthritis (axSpA).

Reviewer 2 Report

Comments and Suggestions for Authors

The manuscript titled: “Increased Prevalence of Headaches and Migraine in Patients with Psoriatic Arthritis and Axial Spondylarthritis: Insights from an Italian Cohort Study” regards the original study analysis of the prevalence of migraines in SpA patients (n=286). Authors have found that the prevalence of migraine without aura was significantly higher in both PsA (18.52%) and axSpA (28.57%) groups than in healthy controls.

1.       Proinflammatory cytokines, similar to SpA, participate in the pathogenesis of migraine. Can migraines be one of the features of PsA or axSpA, or are they co-existing diseases?

2.       How did you treat migraine in patients with PsA and axSpA?

3.       The authors found that fibromyalgia was diagnosed in 38.8% with headache and 61.18% without headache. Why are the differences not statistically important between those groups?

4.       Can fibromyalgia be associated with a higher prevalence of migraine?

5.       The sensation of stimuli is different than in other diseases. How did you diagnose the fibromyalgia? Can fibromyalgia can be a confounding factor in your study?

Author Response

Dear Reviewer,

I sincerely thank you for the comments. We have followed your suggestions and amended our manuscript accordingly. Please find below our responses to the queries raised. We trust these have satisfied your concerns and look forward to hearing from you.

  1. Proinflammatory cytokines, similar to SpA, participate in the pathogenesis of migraine. Can migraines be one of the features of PsA or axSpA, or are they co-existing diseases? Thanks for this comment, as added in the discussion, the common cytokine pattern background between SpA and migraine could explain the genesis and perpetuation of the inflammatory process, since cytokines are important endogenous substances involved in immune and inflammatory responses. (lines 279-290) Furthermore, within the limitations of the study, from line 298 to line 303 we explain that several limitations and strengths apply to the interpretation of our current results. Due to the observational nature of our study, we cannot fully determine causality. Whether the observed effect is caused, in part, by low-grade systemic inflammation caused by arthritis or the findings are primarily the result of shared lifestyle factors and other mechanisms requires further examination. Furthermore, we could not distinguish between the incidence of headache and migraine, whether or not it preceded the development of arthritis.
  2. How did you treat migraine in patients with PsA and axSpA? Thank to the reviewer for the question, we have not treated headaches and migraines in any way in patients with PsA and axSpA, advising in severe cases to access the reference headache center of our hospital.
  3. The authors found that fibromyalgia was diagnosed in 38.8% with headache and 61.18% without headache. Why are the differences not statistically important between those groups? We really apologize for the typo. We insert correct numbers and percentages in table 1: fibromyalgia in entire population 76 (35.68%), in PsA patients with headache 33 (38.82%) and without headache 43 (33.59%).
  4. Can fibromyalgia be associated with a higher prevalence of migraine? Really thank to highlight this topic. We added the following sentences as limitations in the discussion section: We are aware of the high prevalence of fibromyalgia in both populations, likely due to the fact that we used 2016 ACR diagnostic criteria and not clinical parameters such as tender points or self-reported diagnosis. The presence of concurrent fibromyalgia, especially in the context of the headache symptom, which is a hallmark symptom of fibromyalgia, is certainly a limiting factor in our study. Other previous studies have demonstrated the lack of association between comorbid fibromyalgia in rheumatoid arthritis and spondyloarthritis and headache; however, in these studies, the prevalence of fibromyalgia appeared to be underestimated, as it was solely defined as a self-reported diagnosis. We have chosen not to exclude patients with comorbid fibromyalgia from our analysis in order to fully reflect the characteristics of our real-world population, in which fibromyalgia is one of the most prevalent comorbidities in spondyloarthritis.

  5. The sensation of stimuli is different than in other diseases. How did you diagnose the fibromyalgia? Can fibromyalgia can be a confounding factor in your study? We thank the Reviwer for the suggestion. We reported the following sentence in methods section: Patients fulfilling the 2016 American College of Rheumatology revised diagnostic criteria were classified as affected by concomitant fibromyalgia. We also reported these sentences in the discussion, as previously declared: We are aware of the high prevalence of fibromyalgia in both populations, likely due to the fact that we used 2016 ACR diagnostic criteria and not clinical parameters such as tender points or self-reported diagnosis. The presence of concurrent fibromyalgia, especially in the context of the headache symptom, which is a hallmark symptom of fibromyalgia, is certainly a limiting factor in our study. Other previous studies have demonstrated the lack of association between comorbid fibromyalgia in rheumatoid arthritis and spondyloarthritis and headache; however, in these studies, the prevalence of fibromyalgia appeared to be underestimated, as it was solely defined as a self-reported diagnosis. We have chosen not to exclude patients with comorbid fibromyalgia from our analysis in order to fully reflect the characteristics of our real-world population, in which fibromyalgia is one of the most prevalent comorbidities in spondyloarthritis.

Reviewer 3 Report

Comments and Suggestions for Authors

The authors found that PsA and axSpA patients suffer more often with headaches and migraines.

Comments

1.      Introduction: The authors should describe other studies on headache and migraine associated with PsA and axSpA in other world populations.

2.      Line 76: The authors should include here the number of patients in each of the examined groups.

3.      Line 202: The reference is required at the end of this sentence.

4.      Line 202-203: The association of anti-TNFa treatment and headaches was not addressed in the Results section. This should be corrected.

5.      Lines 208-209: The importance of this sentence is not clear. This should be corrected.

6.      Lines 207-213;222-227: The importance of the study is not clear as the authors cannot answer any question related to the association of the headache and migraine role with PsA and axSpA diseases.

Author Response

Dear Reviewer,

I sincerely thank you for the comments. We have followed your suggestions and amended our manuscript accordingly. Please find below our responses to the queries raised. We trust these have satisfied your concerns and look forward to hearing from you.

  1. Introduction: The authors should describe other studies on headache and migraine associated with PsA and axSpA in other world populations. Thank you for your suggestion. In the introduction section we described other cohorts of patients suffering from autoimmune and rheumatological diseases that have been associated with the development of headaches and migraines. In particular from line 70, we reported 3 studies: two concern cohorts of patients suffering from psoriasis, (Sarkhani M et al, An Bras Dermatol. 2023 and Egeberg A et al., J Am Acad Dermatol. 2015 Nov) and a single study analyzing the prevalence of migraine and neuropathic pain in patients suffering from psoriatic arthritis, spondyloarthritis and rheumatoid arthritis (Mathieu S et al, Prevalence of Migraine and Neuropathic Pain in Rheumatic Diseases. J Clin Med. 2020 Jun 17), that show the highest prevalence of migraines observed in patients with PsA, while the highest prevalence of neuropathic pain was found among patients with Spondyloarthritis (SpA)
  2. Line 76: The authors should include here the number of patients in each of the examined groups. We corrected this typo. Thanks for the suggestion. We added the specific number of patients in each of the examined groups: “A single-centre, cross-sectional, observational cohort study has been conducted on 216 PsA patients and 70 axSpA patients.”
  3. Line 202: The reference is required at the end of this sentence. 3. We added reference at the end of the sentence: Mathieu S, Couderc M, Pereira B, Dubost JJ, Malochet-Guinamand S, Tournadre A, Soubrier M, Moisset X. Prevalence of Migraine and Neuropathic Pain in Rheumatic Diseases. J Clin Med. 2020 Jun 17;9(6):1890. doi: 10.3390/jcm9061890. PMID: 32560321; PMCID: PMC7356241.
  4. Line 202-203: The association of anti-TNFa treatment and headaches was not addressed in the Results section. This should be corrected. Thanks for your suggestion. We addressed this topic in the result section, line 189.
  5. Lines 208-209: The importance of this sentence is not clear. This should be corrected. For clarify this sentence we added some studies published so far. From the line 251: In fact, migraine is considered a “brain state” with various alterations in brain network activity influenced by hormonal fluctuations. Estrogen, in particular, has been linked to migraines since the 1970s, and studies have shown that migraines most commonly occur during a rapid decline in estrogen levels in the late luteal phase and early follicular phase, known as hypothesis of estrogen withdrawal. Although withdrawing progesterone does not directly impact migraine onset, it is believed to influence the headache-promoting effects of estrogen fluctuations. The interaction between these hormones and the decrease in GABAergic activity due to progesterone decline in the late luteal phase may be essential in understanding migraine pathophysiology. Additionally, cyclic hormonal fluctuations influence calcitonin gene-related peptide (CGRP) levels, which are higher in women of reproductive age, particularly during periods of elevated estrogen.
  6. Lines 207-213;222-227: The importance of the study is not clear as the authors cannot answer any question related to the association of the headache and migraine role with PsA and axSpA diseases. Thank you for your suggestion. We added two different arguments, derived from other studies, to better clarify the association between headaches and migraine in PsA or axSpA. Firstly, we added some studies published so far. From the line 251: In fact, migraine is considered a “brain state” with various alterations in brain network activity influenced by hormonal fluctuations. Estrogen, in particular, has been linked to migraines since the 1970s, and studies have shown that migraines most commonly occur during a rapid decline in estrogen levels in the late luteal phase and early follicular phase, known as hypothesis of estrogen withdrawal. Although withdrawing progesterone does not directly impact migraine onset, it is believed to influence the headache-promoting effects of estrogen fluctuations. The interaction between these hormones and the decrease in GABAergic activity due to progesterone decline in the late luteal phase may be essential in understanding migraine pathophysiology. Additionally, cyclic hormonal fluctuations influence calcitonin gene-related peptide (CGRP) levels, which are higher in women of reproductive age, particularly during periods of elevated estrogen. On the second hand, we added these sentences in the discussion section: The common pathogenesis of arthritis and migraine may also be related to the vascular and neurogenic components. In fact, it has been demonstrated that the persistence of increased levels of C-reactive protein and high disease activity are predictors of cardio-vascular disease in patients with axial spondyloarthritis [28-32]. Additionally, in murine models of polyarthritis, the density of CGRP-IR fibers is significantly increased and contributes to the generation and maintenance of arthritis pain [33]. Migraine has been associated with an unfavorable cardiovascular risk profile and a 2-fold increased 10-year risk of coronary heart disease. At the same time, CGRP and its canonical receptor, a heterodimer consisting of a calcitonin-like receptor and a protein 1 that modulates receptor activity, are widely expressed in both the central and peripheral nervous systems, as well as in the trigemino-vascular system. They seem to trigger vasodilation and migraine, as demonstrated by plasma levels of CGRP that positively correlate with the intensity and timing of headaches, and endovenous infusion of CGRP that causes migraine-like symptoms in patients with migraine. Moreover,  the main aim of our research was to evaluate the prevalence of headaches and migraine in patients with Psoriatic Arthritis and Axial Spondyloarthritis, and not to demonstrate any relationship and correlation between headache and these rheumatic morbidities.

Round 2

Reviewer 1 Report

Comments and Suggestions for Authors

Dear Authors,

Thank you for incorporating the changes recommended by reviewers' to improve its readability. 

Author Response

We thank the reviewer for accepting our previous modifications to the paper according to their requests.

Reviewer 3 Report

Comments and Suggestions for Authors

The authors made an attempt to improve their paper, however, it requires more work.

Comments

1.      The language of the paper should be improved related to scientific writing. All typos should be corrected.

2.      Lines 162-175 should be moved between lines 137and 138.

3.      Lines 212-213; 289-290: These sentences are not clear. They should be clarified.

4.      Line 219: the term “patients with multiple diseases” is not clear. It should be clarified.

5.      Line 26: Reference is required at the end of this sentence.

6.      Lines 212-219: The authors describe the association of headaches with IBD and estrogen levels but they do not discuss the association of headaches with systemic inflammation in PsA or AxSpA. This should be corrected.

7.      Results: The authors should add Spearman rank correlation analyses to associate headache, migraine, and aura indices with demographic and clinical characteristics of the examined PsA or AxSpA patients.

8.      Discussion should be improved in terms of consistency.

9.      Lines 245-247 vs 251-253: All the repeated statements should be corrected.

Comments on the Quality of English Language

1.      The language of the paper should be improved related to scientific writing. All typos should be corrected.

3.      Lines 212-213; 289-290: These sentences are not clear. They should be clarified.

Author Response

Dear Editor,

I sincerely thank you and the Reviewer for the comments. We have followed his/her suggestions and amended our manuscript accordingly. Please find below our responses to the queries raised. We trust these have satisfied the Reviewers concerns and look forward to hearing from you.

  1. The language of the paper should be improved related to scientific writing. All typos should be corrected. We thank the reviewer and apologize for all the typo. We have corrected the language and grammatical syntax. (i.e. line 79 acknowledgements, line 80 PsA, line 134 in particular, line 152 Charlson comorbidity index, line 259 trigeminal-vascular system, line 261 intravenous).
  2. Lines 162-175 should be moved between lines 137 and 138. We thank the Reviewer for this suggestion. We have moved the text part as suggested.
  3. Lines 212-213; 289-290: These sentences are not clear. They should be clarified. For the lines 212-213 we have changed the sentence as below: In our cohort, patients with headache did not show higher levels of pain, evaluated by patient pain (PP), a tool used by clinicians to evaluate, through a visual analogue scale, the self-reported pain of arthritis patients.

For the line 289-290 we have changed the sentence from the line 289:  In relation to the AxSpA patient cohort, our study did not specifically examine cervical involvement, such as cervical spondylosis or ankylosis, despite the potential for these conditions to trigger headaches. Our assessment did not isolate spinal involvement in distinct regions but rather evaluated it comprehensively.

  1. Line 219: the term “patients with multiple diseases” is not clear. It should be clarified. We apologize for this mistake. This sentence has been removed as it was not pertinent to the discussion. We appreciate the reviewer for pointing this out to us.
  2. Line 26: Reference is required at the end of this sentence. We thank the reviewer for this suggestion, but the line 26 belongs to the abstract and the journal format does not allow citations to be inserted here.
  3. Lines 212-219: The authors describe the association of headaches with IBD and estrogen levels but they do not discuss the association of headaches with systemic inflammation in PsA or AxSpA. This should be corrected. We thank the reviewer for this suggestion; we discussed the association of headaches with systemic inflammation in the second part of discussion, in particular from line 250 to line 268, underlying as the shared pathogenesis of arthritis and migraine may involve vascular and neurogenic components.

The common pathogenesis of arthritis and migraine may also be related to the vascular and neurogenic components. In fact, it has been demonstrated that the persistence of increased levels of C-reactive protein and high disease activity are predictors of cardiovascular disease in patients with axial spondyloarthritis [33]. Additionally, in murine models of polyarthritis, the density of CGRP-IR fibers is significantly increased and contributes to the generation and maintenance of arthritis pain [37]. Migraine has been associated with an unfavorable cardiovascular risk profile and a 2-fold increased 10-year risk of coronary heart disease. At the same time, CGRP and its canonical receptor, a heterodimer consisting of a calcitonin-like receptor and a protein 1 that modulates receptor activity, are widely expressed in both the central and peripheral nervous systems, as well as in the trigemino-vascular system. They seem to trigger vasodilation and migraine, as demonstrated by plasma levels of CGRP that positively correlate with the intensity and timing of headaches, and endovenous infusion of CGRP that causes migraine-like symptoms in patients with migraine [38–40]. The common cytokine pattern background could explain the genesis and perpetuation of this process, as cytokines are important endogenous substances involved in immune and inflammatory responses. Further longitudinal research is required to investigate the role of the spondyloarthritis-migraine comorbidity in defining patients at a very high risk of cardiovascular events.

  1. Results: The authors should add Spearman rank correlation analyses to associate headache, migraine, and aura indices with demographic and clinical characteristics of the examined PsA or AxSpA patients. We thank the reviewer for this suggestion. While the Spearman rank correlation has several advantages, such as not relying on the assumption of normality and being less sensitive to outliers, there are situations where it may not be appropriate or might not provide a complete picture of the relationship between variables, like our data (Spearman's coefficient is appropriate for both continuous and discrete ordinal variables, but headache, migraine, and aura indices are dichotomous variables). For these reasons, we are forced to not be able to carry out a new statistical analysis with Spearman rank correlation.
  2. Discussion should be improved in terms of consistency.  We modified the discussion accordingly and added a paragraph from line 266 to 279: Regarding the use of biological drugs, we found that patients with axSpA who experienced headache actually used anti-TNF-alpha more than patients without an-ti-TNFalpha. This result, which may seem counterintuitive could be a paradoxical effect, with a background similar to what happens in the development of paradox psoriasis [41]. The use of anti-TNF-alpha inhibitors could be linked to the initial use of anti-TNF-alpha drugs, that is started when there is high disease activity, such as may be at the onset of the disease. Furthermore, we know that there are two forms of TNF-alpha: a soluble form, which mainly acts on the TNF type 1 receptor (TNFR1), and a transmembrane form, which acts on the TNF type 2 receptor (TNFR2). TNFR1 binding leads to apoptosis and chronic inflammation; on the other hand, TNFR2 binding promotes cell survival, resolves inflammation, and induces remyelination. TNFR2 is abundant in the central nervous system; therefore, TNF-alpha inhibitors may promote demyelination by blocking the anti-inflammatory and regenerative effects of the transmembrane form of TNF-alpha on TNFR2 and may induce migraine in patients using them [42].
  3. Lines 245-247 vs 251-253: All the repeated statements should be corrected. We thank the reviewer, and we apologize for this typo, we have eliminated the sentence between line 251-253. “In fact, it has been demonstrated that the persistence of increased levels of C-reactive protein and high disease activity are predictors of cardiovascular disease in patients with axial spondyloarthritis [33].”

Round 3

Reviewer 3 Report

Comments and Suggestions for Authors

The authors responded almost all previous Comments.

Comments:

1.      However, the language of the paper was not improved as many sentences remained unclear and scientific writing is still not satisfactory. For example, sentences on Lines 79, 87-91, 214, 217, 221, 226-227, 229, 231, 238, 269-270 should be rephrased.

2.      Line 88: The authors should indicate that 87 participants represent healthy subjects.

3.      Lines 135-136: The number of the examined subjects was not calculated correctly.

4.      Lines 221-224 should be moved to line 267.

Comments on the Quality of English Language

1.      the language of the paper was not improved as many sentences remained unclear and scientific writing is still not satisfactory. For example, sentences on Lines 79, 87-91, 214, 217, 221, 226-227, 229, 231, 238, 269-270 should be rephrased.

Author Response

Dear Editor,

I sincerely thank the Reviewer for the comments. We have followed his/her suggestions and amended our manuscript accordingly. Please find below our responses to the queries raised. We trust these have satisfied the Reviewer concerns and look forward to hearing from you.

  1. However, the language of the paper was not improved as many sentences remained unclear and scientific writing is still not satisfactory. For example, sentences on Lines 79, 87-91, 214, 217, 221, 226-227, 229, 231, 238, 269-270 should be rephrased.

We thank the reviewer for his suggestion. We rephrased as follow:

Line 79: In this setting, our study aimed to evaluate the prevalence of headache, migraine with and without aura, in a cohort of patients with PsA and axSpA followed in a single Italian rheumatologic center and to correlated this datum with demographic and clinical data of the patients. To the best of our knowledge, this represents the first study endeavoring to examine the prevalence of headaches and migraines specifically in Italian participants diagnosed with PsA and axSpA.

Line 87-91: A single-centre, cross-sectional, observational cohort study has been conducted on 216 PsA patients and 70 axSpA patients. At baseline, PsA participants who fulfilled the Classification Criteria for Psoriatic Arthritis (CASPAR) [18] and who were on at least 6 months of follow-up treatment with conventional and/or biologic disease-modifying antirheumatic drugs were considered potentially eligible for the study. For axSpA, we used the New York criteria for ankylosing spondylitis or the axial Assessment of SpondyloArthritis International Society (ASAS) criteria for non-radiographic spondyloarthropathy [19]. Eighty-seven consecutive healthy controls have been enrolled at the Rheumatology Unit, Campus Bio-Medico University, Rome, Italy, from July 2022 to November 2022.

Line 214-217: However, within PsA patients, the incidence of headaches shows a correlation with an increased count of tender and swollen joints, along with higher scores in LEI and HAQ. This indicates that individuals experiencing a more pronounced burden of PsA symptoms are more inclined to develop headaches.

Line 221: As far as axSpA is concerned, we showed that patients who experienced headache used anti-TNF-alpha more frequently. This result may be linked to a more severe disease activity in axSpA patients developing headache compared to those without headache, thus confirming that chronic inflammation may be linked to the onset of headache.

Line 226-227, 229, 231: A noteworthy predominance of females was observed in PsA and axSpA patients with comorbid headache. This phenomenon is rooted in the documented estro-gen-dependent nature of headache; it is widely recognized that variations in estrogen levels, whether increased or decreased, can act as triggers for headaches, including migraines, during different stages of reproductive age such as menstruation, pregnancy, or menopause [30].

Line 238: The interplay between these hormones and the reduction in GABAergic activity stemming from progesterone decrease in the late luteal phase might be pivotal in comprehending migraine pathophysiology. Moreover, cyclical hormonal variations modulate calcitonin gene-related peptide (CGRP) levels, which are elevated in women of reproductive age, especially during heightened estrogen phases.

Line 269-270: Regarding the use of biological drugs, our findings indicate that among patients with axSpA encountering headaches, there was a higher usage of anti-TNF-alpha compared to patients without headaches.  This result, which may seem counterintuitive, could be associated with heightened disease activity in axSpA patients (leading to the adminstration of bDMARDs in these patients) experiencing headaches [41].

  1. Line 88: The authors should indicate that 87 participants represent healthy subjects.

We thank the reviewer. We rephrased as follow: Eighty-seven consecutive healthy controls have been enrolled at the Rheumatology Unit, Campus Bio-Medico University, Rome, Italy, from July 2022 to November 2022.

  1. Lines 135-136: The number of the examined subjects was not calculated correctly.

We apologize for this typo. We rephrased as follow: Two hundred and eightysix participants (216 participants with PsA and 70 with axSpA) and 87 healthy controls were included in the study.

  1. Lines 221-224 should be moved to line 267. We rephrase paragraph at line 267 as follow: Regarding the use of biological drugs, our findings indicate that among patients with axSpA encountering headaches, there was a higher usage of anti-TNF-alpha compared to patients without headaches. This result, which may seem counterintuitive, could be associated with heightened disease activity in axSpA patients (leading to the administration of bDMARDs in these patients) experiencing headaches [41].
